


# Inverse modelling of CF₄ and NF₃ emissions in East Asia

Tim Arnold[1,2,3*], Alistair Manning[1], Jooil Kim[4], Shanlan Li[5], Helen Webster[1], David
Thomson[1], Jens Mühle[4], Ray F. Weiss[4], Sunyoung Park[5,6], and Simon O'Doherty[7]

[1]Met Office, Exeter, UK

[2]National Physical Laboratory, Teddington, Middlesex, UK

[3]School of GeoSciences, University of Edinburgh, Edinburgh, UK

[4]Scripps Institution of Oceanography, University of California, San Diego, La Jolla,
California 92037, USA

[5]Kyungpook Institute of Oceanography, Kyungpook National University, Daegu 41566,
Republic of Korea

[6]Department of Oceanography, Kyungpook National University, Daegu 41566, Republic of
Korea

[7]School of Chemistry, University of Bristol, Bristol, UK.

*Corresponding author tim.arnold@ed.ac.uk

Well mixed abundances and decadal trends of carbon tetrafluoride ($CF_4$) and nitrogen
trifluoride ($NF_3$) have been well characterised and have provided a time series of global total

emissions. Information on locations of emissions contributing to the global total, however, is
currently poor. We use a unique set of measurements between 2008 and 2015 from the Gosan
station, Jeju Island, South Korea (part of the Advanced Global Atmospheric Gases
Experiment network), together with an atmospheric transport model to make spatially
disaggregated emission estimates of these gases in East Asia. Owing to the poor availability

of good prior information for this study our results are strongly constrained by the
atmospheric measurements. Notably, we are able to highlight emissions hotspots of $NF_3$ and
$CF_4$ in South Korea, owing to the measurement location. We calculate emissions of $CF_4$ to be
quite constant between years 2008 and 2015 for both China and South Korea with 2015
emissions calculated at $4.33 \pm 2.65$ Gg yr$^{-1}$ and $0.36 \pm 0.11$ Gg yr$^{-1}$, respectively. Emission



estimates of $NF_3$ from South Korea could be made with relatively small uncertainty at $0.6 \pm$ 0.07 Gg $yr^{-1}$ in 2015, which equates to ~1.6% of the country's $CO_2$ emissions. We also apply our method to calculate emissions of $CHF_3$ (HFC-23) between 2008 and 2012, for which our results find good agreement with other studies and which helps support our choice in methodology for $CF_4$ and $NF_3$.

**1.  Introduction**

The major greenhouse gases (GHGs) − carbon dioxide, methane and nitrous oxide − have natural and anthropogenic sources. The synthetic fluorinated species (chlorofluorocarbons, hydrochlorofluorocarbons, hydrofluorocarbons, and perfluorocarbons) are almost or entirely anthropogenic and are released from industrial and domestic appliances and applications. Of

the synthetic species, tetrafluoromethane ($CF_4$) and nitrogen trifluoride ($NF_3$) are emitted nearly exclusively from point sources of specialized industries (Arnold et al., 2013; Mühle et al., 2010). Although these species currently make up only a small percentage of current emissions contributing to global radiative forcing, they have potential to form large portions of specific company, sector, state, province, or even country level carbon budgets.

$CF_4$ is the longest-lived GHG gas known with an estimated lifetime of 50,000 years, leading to a global warming potential on a 100-year time scale ($GWP_{100}$) of 6630 (Myhre et al., 2013). Significant increases in atmospheric concentrations are ascribed mainly to emissions from primary aluminum production during so-called "anode events" when the alumina feed to the reduction cell is restricted (International Aluminium Institute, 2016), and from the

microchip-manufacturing component of the semiconductor industry (Illuzzi and Thewissen, 2010). Recently, evidence is emerging that, similar to primary aluminium production, rare earth element production may also release substantial amounts of $CF_4$ (Vogel et al., 2017; Zhang et al., 2017). Other emission sources for $CF_4$ include release during the production of $SF_6$ and HCFC-22, but emissions from these sources are estimated to be small compared to

the emissions from the aluminium production and semiconductor manufacturing industries (EC-JRC/PBL, 2013; Mühle et al., 2010). There is also a very small natural emission source of $CF_4$, sufficient to maintain the preindustrial atmospheric burden (Deeds et al., 2008).

According to the IPCC fifth assessment, $NF_3$'s global warming potential on a 100-year time scale ($GWP_{100}$) is ∼16,100 (based on an atmospheric lifetime of 500 years) (Myhre et al.,

2013), however, recent work suggests the $GWP_{100}$ is higher at 19,700 (Totterdill et al., 2016). Use of $NF_3$ began in the 1960s in specialty applications, e.g., as a rocket fuel oxidizer



and as a fluorine donor for chemical lasers (Bronfin and Hazlett, 1966). More recently, beginning in the late 1990s, $NF_3$ has been used by the semiconductor industry, and in the production of photovoltaic cells and flat-panel displays. $NF_3$ can be broken down into

reactive fluorine (F) radicals and ions, which are used to remove the remaining silicon-containing contaminants in process chambers (Henderson and Woytek, 1994; Johnson et al., 2000). $NF_3$ was also chosen because of its promise as an environmentally friendly alternative, with conversion efficiencies (to create reactive F) far higher than other compounds such as $C_2F_6$ (Johnson et al., 2000; International SEMATECH Manufacturing Initiative, 2005). Given

its rapid recent rise in the global atmosphere and projected future market, it has been estimated that $NF_3$ could become the fastest growing contributor to radiative forcing of all the synthetic GHGs by 2050 (Rigby et al., 2014b).

$CF_4$ and $NF_3$ are not the only species with major point source emissions. Trifluoromethane ($CHF_3$; HFC-23) is principally made as a byproduct in the production of

chlorodifluoromethane ($CHClF_2$, HCFC-22). Of the hydrofluorocarbons (HFCs), HFC-23 has the highest 100-year global warming potential ($GWP_{100}$) at 12,400 owing most significantly to a long atmospheric lifetime of 222 years (Myhre et al., 2013). Its regional and global emissions have been the subject of numerous previous studies (Fang et al., 2014; McCulloch and Lindley, 2007; Miller et al., 2010; Montzka et al., 2010; Fang et al., 2015; Stohl et al.,

2010; Li et al., 2011; Kim et al., 2010; Yao et al., 2012; Keller et al., 2012; Yokouchi et al., 2006). Thus, emissions of HFC-23 are already relatively well characterized from a bottom up and top-down perspective. In this work we will also calculate HFC-23 emissions, not to add to current knowledge, but to provide a level of confidence for our methodology.

Unlike for HFC-23, the spatial distribution of emissions responsible for $CF_4$ and $NF_3$

abundances is very poorly understood, which is hindering action for targeting mitigation. HFC-23 is emitted from well-known sources (namely HCFC-22 production areas) with well characterized estimates of emission magnitudes and hence it has been a target for successful mitigation (by thermal destruction) via the clean development mechanism (Miller et al., 2010). However, emissions of $CF_4$ and $NF_3$ are very difficult to estimate from industry level

information: Emissions from Al production are highly variable depending on the conditions of manufacturing and emissions from the electronics industry depend on what is being manufactured and individual companies' recipes for production (such information is not publicly available). Both the Al production and semiconductor industries have launched voluntary efforts to control their emissions of these substances, reporting success in meeting





their goals (International Aluminium Institute, 2016; Illuzzi and Thewissen, 2010; World
Semiconductor Council, 2017). Despite the industry's efforts to reduce emissions, top-down
studies on the emissions of $CF_4$ and $NF_3$ have shown the bottom-up inventories are likely to
be highly inaccurate. Most recently, Kim et al. (2014) showed that global bottom-up
estimates for $CF_4$ are as much as 50% lower than top-down estimates, and Arnold et al.

(2013) show that the best estimates of global $NF_3$ emissions calculated from industry
information and statistical data total only ~35% of that estimated from atmospheric
measurements.

       Accurate emission estimates of $NF_3$ and $CF_4$ are difficult to make based on simple parameters
such as integrated country level uptake rates and leakage rates, which, for example, underpin

calculations of HFC emissions. Active or passive activities to reduce emissions vary between
countries, and between industries and companies within countries, and the impetus to
accurately understand emissions is lacking in regions that have not been required to report
emissions under the UNFCCC. This problem is compounded by the difficulty in making
measurements of these gases: $CF_4$ and $NF_3$ are the two most volatile GHGs after methane

($CH_4$) with low atmospheric abundances, which makes routine measurements in the field at
the required precision particularly difficult. The Advanced Global Atmospheric Gases
Experiment (AGAGE) has been monitoring the global atmospheric trace gas budget for
decades (Prinn et al., 2018). Most recently, AGAGE's 'Medusa' pre-concentration GC-MS
(gas chromatography-mass spectrometry) system has been able measure a full suite of the

long-lived halogenated GHGs (Arnold et al., 2012; Miller et al., 2008). The Medusa is the
only instrument demonstrated to measure $NF_3$ in ambient air samples, and the only field-
deployable instrument capable of measuring $CF_4$. The Medusa on Jeju Island is one of only
twenty such instruments currently in operation globally and is uniquely sensitive to the
dominant emission sources of these compounds given its location in this highly industrial part

of the globe with large capacities of Al production, semiconductor manufacturing, and rare
earth element production industries. Its utility has already been demonstrated in numerous
previous studies to understand emissions of many GHGs from Japan, South Korea, North
Korea, eastern China, and surrounding countries (Fang et al., 2015; Kim et al., 2010; Li et al.,
2011).

For the first time we use the measurements of $CF_4$ (starting in 2008) and $NF_3$ (starting in
2013) in an inversion framework – coupling each measurement with an air history map
computed using a particle dispersion model. We demonstrate the use of these measurements



to find emissions hotspots in this unique region with minimal use of prior information, and
we show that East Asia is a major source of these species. Focussed mitigation efforts, based

on these results, could have a significant impact on reducing GHG emissions from specific
areas. The technology for abating emissions of these gases from such discrete sources exists
and could be used (Chang and Chang, 2006; Purohit and Höglund-Isaksson, 2017; Illuzzi and
Thewissen, 2010; Yang et al., 2009; Raoux, 2007; Wangxing et al., 2016).

## 2.  Methods

### 2.1 Atmospheric measurements

The Gosan station (from here termed GSN) is located on the south-western tip of Jeju Island
in the Republic of Korea (126.16181° E, 33.29244° N). The station rests at the top of a 72 m
cliff, about 100 km south of the Korean peninsula, 500 km northeast of Shanghai, China, and

250 km west of Kyushu, Japan, with an air inlet 17 m above ground level.

A Medusa GC-MS system was installed at GSN in 2007 and has been operated as part of the
AGAGE network to take automated, high-precision measurements for a wide range of CFCs,
HCFCs, HFCs, PFCs, Halons and other halocarbons; all significant synthetic GHG and/or
stratospheric ozone depleting gases as well as many naturally occurring halogenated

compounds (Miller et al., 2008; Arnold et al., 2012; Kim et al., 2010). Since November 2013,
$NF_3$ has been measured within this suite of gases. Air reaches GSN from the most heavily
developed areas of East Asia, making the measurements and their interpretation a unique
source for 'top-down' emissions estimates in the region. Ambient air measurements are made
every 130 minutes and are bracketed with a standard before and after the air sample to correct

for instrumental drift in calibration. Further details on methodology for calibration of these
gases is given elsewhere (Arnold et al., 2012; Mühle et al., 2010; Miller et al., 2010; Prinn et
al., 2018).

### 2.2 Atmospheric model

Lagrangian particle dispersion models are well suited to this type of study as they can be run

backwards allowing for the source-receptor relationship to be efficiently calculated. We use
the Numerical Atmospheric dispersion Modelling Environment (NAME III), henceforth
called NAME, developed by the UK Met Office (Ryall and Maryon, 1998; Jones et al.,
2007), with meteorological parameter inputs from the Met Office's operational global NWP
model, the Unified Model (UM) (Cullen, 1993). The UM has a horizontal resolution of



0.5625 × 0.375 ° (~40 km) from December 2007 to April 2010; 0.3516 × 0.2344 ° (~25 km)
        from April 2010 to July 2014; and 0.234375 × 0.15625 ° (~17 km) from mid-July 2014 to
        mid-July 2017. The number of vertical levels in the UM has increased over this period, with
        NAME taking the lowest 31 levels in 2009 and the lowest 59 levels in 2015. The GHGs
        considered in this study have lifetimes on the order of hundreds to tens of thousands of years

(Myhre et al., 2013), and can be considered inert gases on the spatial and temporal scales of
        this study and therefore the NAME model schemes for representing chemistry, dry
        deposition, wet deposition and radioactive decay were not used. The boundary layer height
        (BLH) estimates are taken from the UM, however, a minimum BLH allowed within NAME
        was set to 40 m to be consistent with the maximum emission height and the height of the

output grid. The NAME model was run to estimate the 30-day history of the air on route to
        GSN. The NAME model output estimates the time-integrated air concentration (dosage) at
        each grid box (0.352° × 0.234°, and 0–40 m above ground level, irrespective of the
        underlying UM meteorology resolution) from a release of 1 g s$^{-1}$ at GSN at 10±10 metres
        above the model ground level (magl). The model is three-dimensional therefore it is not just

surface to surface transport that is modelled: An air parcel can travel from the surface to a
        high altitude and then back to the surface but only those times when the air parcel is within
        the lowest 40 m above the ground will it be included in the model output aggregated
        sensitivity maps. The computational domain covers 54.34° E to 168.028° W longitude (391
        grid cells of dimension 0.352°) and 5.3°S to 74.26°N latitude (340 grid cells of dimension

0.234°) and extends to more than 19 km vertically. Despite the increase in the resolution of
        the UM over the time period covered, the resolution of the NAME output was kept constant
        throughout.  For each 1 h period, 5000 inert model particles were used to describe the
        dispersion of air. By dividing the dosage [g s m$^{-3}$] by the total mass emitted [3600 s h$^{-1}$ × 1 h
        × 1 g s$^{-1}$] and multiplying by the geographical area of each grid box [m$^2$], the model output

was converted into a dilution matrix D [s m$^{-1}$]. Each element of this matrix D dilutes a
        continuous emission of 1 g m$^{-2}$ s$^{-1}$ from a given grid box over the previous 30 d to simulate
        an average concentration [g m$^{-3}$] at the receptor (measurement point) during a 1 h period.

        ### 2.3 Inversion framework

For most long-lived trace gases (with lifetimes of years or longer), the assumption that
        atmospheric mole fractions respond linearly to changes in emissions holds well. By using this
        linearity, we can relate a vector of observations ($y$) to a state vector ($x$) made up of emissions





and other non-prescribed model conditions (see section 2.6), via a sensitivity matrix ($H$) (Tarantola, 2005):

$$y = Hx + residual \qquad (1)$$

A Bayesian framework is typically used in trace gas inversions and incorporates a priori information, which gives rise to the following cost function:

$$C = (Hx - y)^T R^{-1} (Hx - y) + (x - x_p)^T B^{-1} (x - x_p) \qquad (2)$$

Where, $C$ is the cost function score (the aim is to minimise this score); $H$ is made up mainly of the model derived dilution matrices (Section 2.2) but also the sensitivity of changes in boundary conditions to measured mixing ratios; $x$ is a vector of emissions and boundary conditions; $y$ is a vector of observations; $R$ is a matrix of combined model and observation uncertainties; $x_p$ is a vector of prior estimates of emissions and boundary conditions; and $B$ is an error matrix associated with $x_p$. The cost function is minimised using a non-negative least squares fit (Lawson and Hanson, 1974), as done previously for volcanic ash (Thomson et al., 2017; Webster et al., 2017).

The first term in equation 2 describes the mismatch (fit) between the modelled time-series and the observed time-series at each observation station. The observed concentrations ($y$) are comprised of two distinct components; (a) the Northern Hemisphere background concentration, referred to as the baseline, that changes only slowly over time, and (b) rapidly varying perturbations above the baseline. These observed deviations above background (baseline) are assumed to be caused by emissions on a regional scale that have yet to be fully mixed on the hemisphere scale. The magnitude of these deviations from baseline and, crucially, how they change as the air arriving at the stations travels over different areas, is the key to understanding where the emissions have occurred. The inversion system considers all of these changes in the magnitude of the deviations from baseline as it searches for the best match between the observations and the modelled time-series. The second term describes the mismatch (fit) between the estimated emissions and boundary conditions ($x$) and prior estimated emissions and boundary conditions ($x_p$) considering the associated uncertainties ($B$).

The aim of the inversion method is to estimate the spatial distribution of emissions across a defined geographical area. The emissions are assumed to be constant in time over the inversion time period (in this case 1 calendar year as is typically reported in inventories). Assuming the emissions are invariant over long periods of time is a simplification, but is necessary given the





limited number of observations available. In order to compare the measurements and the model
time-series, the latter are converted from air concentration [g m$^{-3}$] to the measured mole
fraction (e.g. parts per trillion [ppt]) using the modelled temperature and pressure at the
observation point.

### 2.4 Prior emissions information

Global emissions estimates of $CF_4$ and $NF_3$ using atmospheric measurements have
demonstrated that 'bottom-up' accounting methods for one or more sectors, or one or more
regions, are highly inaccurate (Arnold et al., 2013; Mühle et al., 2010). This study makes no
effort to improve such inventory methods but instead focuses on minimising the reliance of
prior information on our Bayesian-based posterior emissions estimates. Our prior information
data sets come from the EDGAR (emissions database for atmospheric research) v4.2
emission grid maps (EC-JRC/PBL, 2013). This data set only covers the years 2000 to 2010
and therefore we apply the prior for 2010 for each year between 2011 and 2015. The 0.1 x
0.1° EDGAR emission maps were first re-gridded based on the lower resolution of our
inversion grid (0.3516° × 0.2344°). In order to remove the influence of the within-country
prior spatial emissions distribution, each country's emissions were then averaged across their
entire landmass (see Figure S1). We applied 5 different levels of uncertainty to each
inversion grid cell (a,b) in 5 separate inversion experiments, each a multiple of the emissions
magnitude ($x_{a,b}$) in each grid cell: $1 \times x_{a,b}$ (i.e. 100% uncertainty), $10 \times x_{a,b}$, $100 \times x_{a,b}$, $1000 \times x_{a,b}$,
and $10,000 \times x_{a,b}$. We were then able to test the sensitivity of the prior emissions uncertainty
and provide evidence for the low influence of prior information on the emissions estimates in
the posterior.

### 2.5 Model-measurement and prior uncertainties

As well as inaccurate prior information, another significant source of uncertainty in estimating
emissions is through the modelling; both the underlying meteorology and the atmospheric
transport model itself. The uncertainty matrix, $R$, is a critical part of equation 2 that allows us
to adjust uncertainties assigned to each measurement depending on how well we think the
model is performing at that time: It describes, per hour time period, a combined uncertainty of
the model and the observation at each time. The method of assigning measurement-model
uncertainties is under development and here we describe one method that has been applied to
the modelling of GSN measurements. Emissions are primarily diluted by transport and mixing
within the planetary boundary layer (PBL), and hence, modelling of the PBL height (BLH) is



crucial for accurate modelling of the mixing ratios. To assign a model uncertainty to each hourly window of measurements we use model information of BLH, as well as how well the model is representing the topography and the inlet height at the measurement site:

$$\sigma_{model} = \sigma_{baseline} \times f_{BLH} \times f_{Topography} \times f_{Inlet} \quad (3)$$

where, $\sigma_{baseline}$ is the variability associated with the baseline calculation (see Section 2.6), and $f_{BLH}$, $f_{Topography}$, and $f_{Inlet}$ are multiplying factors (greater than or less than unity) that increase or decrease the relative uncertainty assigned to each model time period. $f_{BLH}$ is based on boundary layer height magnitude and variability over a three-hour period; $f_{Topography}$ is the mismatch between the modelled and actual altitude of the site (above sea level); and $f_{Inlet}$

is the mismatch between the modelled and actual inlet heights. Differences between model and actual topography surrounding the coastal site of GSN are relatively small, as is the mismatch between inlet height, and therefore $f_{Topography}$ and $f_{Inlet}$ can be considered as having a negligible impact of the overall model uncertainty in this study i.e. $f_{Topography}$ and $f_{Inlet}$ equal one. Thus, $f_{BLH}$ largely dictates the relative model uncertainty assigned to each

observation.

If the model boundary layer height transitions across the sample inlet height (i.e. it cannot be confidently known whether sampling was from above or below the boundary layer) then $f_{BLH}$ is given a very large value (arbitrarily set at 10,000), which effectively removes that observation from consideration in the inversion. Otherwise $f_{BLH}$ is calculated with the

following:

$$f_{BLH} = \frac{Max_{BLH-inlet}}{Min_{BLH-inlet}} \times \frac{Threshold}{Min_{BLH}}$$

where, $Max_{BLH-inlet}$ is the largest of either 100 m or one of the distances calculated hourly between the inlet and the modelled BLH within a period of three hours around the measurement time; $Min_{BLH-inlet}$ is the smallest of the distances calculated between the inlet

and the BLH over the same three-hour period; $Threshold$ is an arbitrary value set at 500 m; and $Min_{BLH}$ is the lowest boundary layer height recorded over the three-hour period. Thus, the relative assigned uncertainty takes into account the proximity of the varying boundary layer to the inlet height and a recognition that observations taken when the boundary layer is varying at higher altitudes (>500 m a.g.l.) is likely to be better modelled and therefore have

lower uncertainty compared to those taken when the BLH is varying at lower altitudes (< 500 m a.g.l.).



Supporting Figures S2-S6 show annual time series of observations and the corresponding measurement-model uncertainties, as well as statistics for the mismatch between observations and modelled time series.

**2.6 Baseline calculation and boundary conditions**

For each measurement at GSN it is important to accurately understand the portion of the total mixing ratio arriving from outside the inversion domain and the portion from emission sources within the domain, otherwise emissions from specific areas could be significantly over or under estimated. GSN is uniquely situated; receiving air masses from all directions

over the course of the year, which can have distinct compositions of trace gases, driven mainly by the different emission rates between the two hemispheres and slow inter-hemispheric mixing.

In addition to the time integrated air concentration produced by NAME (Section 2.2), the 3D coordinate where each particle left the computational domain was also recorded. This

information was then post-processed to produce the percentage contributions from 11 different borders of the 3D domain (Figure 1). From 0 to 6 km in height eight horizontal boundaries (WSW, WNW, NNW, NNE, ENE, ESE, SSE, SSW) were considered and between 6 to 9 km the horizontal boundaries were only split between north and south. The eleventh border was considered when particles left in any direction above 9 km. Thus, the

influence of air arriving to GSN from outside the domain was simplified as a combination of air masses arriving from 11 discrete directions.

We use measurements from the Mace Head observatory (from here termed MHD) on the west coast of Ireland (53.33° N, 9.90° W) – a key AGAGE (Advanced Global Atmospheric Gases Experiment) site providing long term in-situ atmospheric measurements – to act as a

starting point for an estimate of the composition of air from the mid-latitudes entering the East Asia domain. MHD was one of the first locations to measure $CF_4$ (starting 2004) and $NF_3$ (starting 2012) and other measurements from the site are routinely used in atmospheric studies to calculate the time-varying well-mixed mid-latitude Northern Hemisphere abundances. In summary, a quadratic fit was made only to MHD observations that were

representative of the NH baseline i.e. when well mixed air was arriving predominately from the WNW-NNW (North Atlantic) direction as calculated using NAME (details of filtering and fitting are given in the Supporting Text).



The composition of air arriving from any of the 11 directions is calculated using corresponding multiplying factors applied to the MH baseline, which were included as part of

the state vector $x$ in Equation 2. The prior baseline was therefore perturbed as part of the inversion based on the relative contribution of air arriving from different borders of the 3D domain and the multiplying factors are included within the cost function (Equation 2). Figure 2 shows an annual time series of observations for CF$_4$ and the difference between the prior baseline (the quadratic fit from MHD) and the posterior baseline.

**2.7 Domains and inversion grids**

The domain used in the inversion is smaller than the computational NAME transport model domain; The horizontal inversion domain covers 88.132° E to 145.860° E longitude (164 fine grid cells of 0.352°) and 15.994° N to 57.646° N latitude (178 fine grid cells of 0.234°). As GSN is surrounded by highly industrial areas the site is insensitive to small emissions or

highly diluted emissions from further away. NAME is run on a larger domain to ensure that on the occasion when air circulates out of the inversion domain and then back, its full 30-day history in the inversion domain is included. Further, although the very large area for our model domain may not be necessary for this study, the model will not need to be run again should a larger area of analysis be required.

An initial computational inversion grid (from here termed the 'coarse grid') was created based on aggregated information from the NAME footprints over the period of the inversion (in this case one year), aggregating fewer grid cells in areas that are 'seen' the most by GSN, and on the prior emissions flux i.e. areas known to have low emissions (i.e. the ocean) had higher aggregation.  Course grid cells could not be aggregated over more than a single

country and a total of ≈100 coarse grid cells (n) were created. After the initial inversion the posterior emissions density [g yr$^{-1}$ m$^{-2}$] and the ability of the posterior emissions to impact the measurements at GSN (using information from the NAME output) was used to choose a coarse grid cell to divide in two by area. A new inversion was run using identical inputs except for the number of grid cells (now n+1). This sequence was repeated 50 times creating

≈150 coarse grid cells within the inversion domain.

### 3.   Results and discussion

**3.1 Country total emissions estimates**

Table 1 provides a summary of our estimates of emissions from the five major emitting countries/regions within the East Asia domain. These posterior emission estimates use a prior





emissions uncertainty in each fine grid cell of 100x the emissions magnitude (see Section
2.4).

*HFC-23*

Fang et al. (2015) conducted a very thorough 'bottom-up' study within their work on HFC-
23; constraining an inversion model using both prior information and atmospheric

measurements. They used an inverse method based on FLEXPART using measurements
from three sites in East Asia – GSN, Hateruma (a Japanese island ~200 km east of Taiwan),
and Cape Ochi-ishi (northern Japan), calculating an HFC-23 emissions rise in China from 6.4
$\pm$ 0.7 Gg yr$^{-1}$ in 2007 (6.2 $\pm$ 0.6 Gg yr$^{-1}$ in 2008) to 8.8 $\pm$ 0.8 Gg yr$^{-1}$ in 2012. An earlier study
by Stohl et al. (2010) also report HFC-23 emissions of 6.2 $\pm$ 0.8 Gg yr$^{-1}$ in 2008. Both Fang

et al. (2015) and Stohl et al. (2010) report emissions from other countries below 0.25 Gg yr$^{-1}$
for all years. Our estimates use a completely independent inverse method and only data from
GSN, yet the results are very close to those from Fang et al. (2015) (Figure 3): 6.8 $\pm$ 4.3 Gg
yr$^{-1}$ in 2008 (a difference of 10%) and 10.7 $\pm$ 4.6 Gg yr$^{-1}$ in 2012 (a difference of 22%), and
to Stohl et al. (2015). For estimates of emissions from other countries our posterior estimates

are not constrained sufficiently to make a meaningful comparison with Fang et al. (2015).
The posterior uncertainties in these two different studies mainly reflect the difference in the
prior uncertainty assumed for the prior information: We assume a very poor level of certainty
on our prior emissions and therefore our posterior uncertainties are a lot higher. However,
these inversion results are lower than estimates based on inter-species correlation analysis by

Li et al. (2011) who calculated emissions of HFC-23 from China in 2008 in the range of 7.2-
13 Gg yr$^{-1}$. And using a CO tracer-ratio method, Yao et al. (2012) estimated particularly low
emissions of 2.1 $\pm$ 4.6 Gg yr$^{-1}$ for 2011-2012. The estimates derived from atmospheric
inversions do not rely on any correlations with other species or known emissions for certain
species, and given two separate inversion studies have produced very similar results we

suggest these provide a more reliable 'top-down' emissions estimate of HFC-23. As well as
providing an independent validation of the previous work on HFC-23 by Fang et al. (2015)
and Stohl et al. (2010), the alignment of our HFC-23 emissions estimates with those previous
studies provides confidence in our inversion methodology for the CF$_4$ and NF$_3$ emissions
estimates.

*CF$_4$*





Our understanding of emissions of $CF_4$ and $NF_3$ is very poor, which is highlighted in global studies based on atmospheric measurements that show bottom-up estimates of emissions are significantly underestimated (Mühle et al., 2010; Arnold et al., 2013). With such a poor prior understanding of emissions we assess the effect of prior uncertainty on the posterior

emissions (Figure 3). With assignment of uncertainty on the prior of each fine grid cell at ten times the prior emissions value, the posterior is still significantly constrained by the prior for both China and South Korea. Posterior estimates when larger uncertainties are applied to the prior (100x to 10000x) are very consistent, indicating that when greater than 100x uncertainty is applied, emissions estimates are most significantly constrained by the atmospheric

measurements. For China our posterior estimates are greater than twice the prior estimates taken from EDGAR v4.2. Mühle et al. (2010) estimated global $CF_4$ emissions at $10.5 \pm 0.4$ Gg yr$^{-1}$ between 2005 and 2008 and our estimate for Chinese emissions in 2008 of $4.66 \pm 1.82$ Gg yr$^{-1}$ suggests ~44 % of total global $CF_4$ emissions come from China. This is significantly higher than emission estimation methods using interspecies correlation: Kim et al. (2010)

estimated $CF_4$ emissions in the range of only 1.7-3.1 Gg yr$^{-1}$ in 2008 and Li et al. (2011) only 1.4-2.9 Gg yr$^{-1}$ over the same period. The interspecies correlation approach inherently requires that the sources of the different gases that are compared are coincident in time and space. Kim et al. (2010) and Li et al. (2011) used HCFC-22 as the tracer compound for China with a calculated emissions field from an inverse model and most emissions of this gas

originate from fugitive release from air conditioners and refrigerators. However, $CF_4$ is emitted mostly from point sources in the semiconductor and aluminium production industries with different spatial emissions distribution within countries and likely different temporal characteristics compared to HCFC-22.

Emissions estimates from South Korea and Japan are an order of magnitude lower than for

China. For 2008 Li et al. (2011) estimate emissions of $CF_4$ from combined South and North Korea of 0.19-0.26 Gg yr$^{-1}$ and from Japan of 0.2-0.3 Gg yr$^{-1}$, which are on the low end of the uncertainty range of our estimates for that year (Table 1). As one of the largest, if not the largest, country for semiconductor wafer production, Taiwan is also an emitter of $CF_4$. However, measurements at GSN provide only poor sensitivity to detection of emissions from

Taiwan and our results can only suggest that emissions are likely <0.5 Gg yr$^{-1}$. North Korea emissions were small and no annual estimate was above 0.1 Gg yr$^{-1}$.

*$NF_3$*





Our understanding of $NF_3$ emissions from inventory and industry data is even poorer than for $CF_4$. On a global scale the emission estimates from industry are underestimated (Arnold et

al., 2013). This study suggests that at least some emissions of $NF_3$ stem from China, however gaining meaningful quantitative estimates has been difficult due to large uncertainties (Figure 3). Contrastingly, the posterior estimates of emissions from South Korea have relatively small uncertainties. Emissions from China travel a greater distance to the measurement site compared to emissions from South Korea. Thus, the magnitude of $NF_3$ pollution events from

China (especially from provinces furthest west), in terms of the mixing ratio detected at GSN, are smaller than for pollution arriving from neighbouring South Korea. Also, the poorer measurement precision for $NF_3$ compared to $CF_4$ leads to slightly poorer certainty on the baseline, which in turn affects the certainty on the pollution episode, especially for more dilute signals. Emissions from Japan are likely significant (as with Chinese emissions) but

more precise estimates are difficult to make without improved prior information and more atmospheric measurements in other locations. As for $CF_4$, emission estimates of $NF_3$ from Taiwan and North Korea are highly uncertain. However, our results do indicate that emissions of $NF_3$ from Taiwan might be lower than from South Korea despite very similar sized semiconductor production industries. Focussing on the more meaningful estimates from

South Korea, emissions of $NF_3$ in 2015 are estimated to be $0.6 \pm 0.07$ Gg $yr^{-1}$ which equates to $9660 \pm 1127$ Gg $yr^{-1}$ $CO_2$-equivalent emissions (based on a $GWP_{100}$ of 16,100). This is ~1.6% of the country's $CO_2$ emissions (Olivier et al., 2017), thus making a significant impact on their total carbon budget. Further, given that the sources of $NF_3$ are relatively few, these emissions can be assigned to a small number of industries, potentially making $NF_3$ an easy

target for focussed mitigation policy. Rigby et al. (2014a) updated the global emission estimates from Arnold et al. (2013), calculated an annual emissions estimate of 1.61 Gg $yr^{-1}$ for 2012, with an average annual growth rate over the previous 5 years of 0.18 Gg $yr^{-1}$. Linearly extrapolating this growth to 2014 and 2015 leads to projected global emissions of 1.97 and 2.15 Gg $yr^{-1}$ for 2014 and 2015, respectively. Thus, South Korean emissions as a

percentage of these global totals equate to ~20% and ~28% for 2014 and 2015, respectively, which is around the proportion of semiconductor wafer fabrication capacity in South Korea relative to global totals (~20%) (SEMI, 2017).

### 3.2 Spatial emission maps



Figure 4 shows the effect of regridding over the course of 50 separate $CF_4$ inversions (for 2015), from zero regridding steps (i.e. using a coarse grid space determined using information from NAME and the prior emissions), through to 25, and then 50 steps. The inversion was not allowed to decrease the minimum posterior grid size beyond four fine grid squares (i.e. four times the $0.3516° \times 0.2344°$ grid square). This method highlights the areas

that have the highest emissions density; the splitting of these grid cells improves the correlation between observations and posterior model output. However, these emission maps must be studied alongside the corresponding uncertainty maps. The inversion could continue to split towards a fine grid resolution limit, however, there may not be enough information in the data to accurately constrain emissions from each course grid cell (leading to spurious

emission patterns) and the process would be computationally very expensive. The largest emissions of $CF_4$ arise from China and Figure 4 suggests the largest emissions come from an area between $35°$ N and $38°$ N. The uncertainty on these emissions from the specific final coarse grid squares is large (Figure 4F) and therefore care needs to be taken not to over interpret emission hotspots. Although the grid is being split it is not realistic for the model to

correctly interpret the spatial distribution of emissions at this distance from GSN, and this is demonstrated in Figure 4F where the relative error on emissions in this corner of the domain is large. Without better prior information it is not possible to distinguish between real year-to-year emission pattern changes and inaccurate emission patterns (Figure 5 and S7). Over the period of study emissions of $CF_4$ generally appear to arise from north of $30°$ and in 2008 and

2013 emissions appear around $25°$ N. However, GSN does not have good sensitivity to emissions from this area and it is possible that these emissions could be incorrectly assigned from Taiwan. Although emissions from South Korea are significantly lower than for China, the proximity to GSN causes the grid cells to be split and emissions to be assigned at higher spatial resolution, and generally (except for 2008) in the north-west quadrant of the country.

Splitting of grid cells in South Korea decreased the relative error on the emissions from particular grid squares, providing confidence that the placement of emissions is accurate. Further, for sequential years 2013, 2014 and 2015 two specific grid cells in that north-west quadrant are highlighted with comparatively low uncertainties (Figure S7). How well these consistent year-to-year emission patterns in South Korea correlate with the actual location of

emissions needs to be the subject of further study (e.g. improved bottom-up inventory compilation efforts). Emissions from Japan are too uncertain to explore the spatial emissions pattern.



For NF$_3$, emissions from China and Japan are too low and uncertain to interpret at finer spatial resolution. However, as with CF$_4$, it is interesting to study the relatively more certain

spatially disaggregated emissions from South Korea (Figure 6). In common with CF$_4$, NF$_3$ emissions from the south-west area are minimal, however in contrast to CF$_4$, emissions occur on the eastern side of South Korea and on the south east coast. Emissions from the south east coast coincide with the known location of a production plant for NF$_3$ located in the area of Ulsan (Gas World, 2011). If this plant is sufficiently separated in space from the end-users of

NF$_3$ then this result would indicate that production of NF$_3$, not just use, could be a significant source in South Korea.

The study of Fang et al. (2015) highlights three major hotspots for HFC-23 emissions in China based on HCFC-22 production facility locations. Our posterior maps correctly show the bulk of emissions in far east China, in line with the results of Fang et al. (2015).

However, given the inconsistency of emissions maps between years we are unable to provide any more information without a better spatially disaggregated prior emissions map.

**Conclusions**

We largely remove the influence of prior 'bottom-up' information and present the first

Bayesian inversion estimates of CF$_4$ and NF$_3$ from the East Asia region using measurements from a single atmospheric monitoring site, GSN station located on the island of Jeju (South Korea). The largest CF$_4$ emissions are from China, estimated at 4-6 Gg yr$^{-1}$ for six out of the eight years studied, which is significantly larger than previous estimates. Despite significantly smaller emissions from South Korea, the spatial disaggregation of CF$_4$

emissions were consistent between independent inversions based on annual measurement data sets, indicating the north west of South Korea is a hotspot for significant CF$_4$ release, presumably from the semiconductor industry. Emissions of NF$_3$ from South Korea were quantifiable with significant certainty, and represent large emissions on a CO$_2$-equivalent basis (~1.6% of South Korea's CO$_2$ emissions in 2015). HFC-23 emissions were also

calculated using the same inversion methodology with high uncertainty on prior information. We found good agreement with other studies in terms of aggregated country totals and spatial emissions patterns, providing confidence that our methodology is suitable and conclusions justified for estimates of CF$_4$ and NF$_3$.



Our results highlight an inadequacy in both the bottom-up reported estimates for $CF_4$ and $NF_3$ and the limitations of the current measurement infrastructure for 'top-down' estimates for these specific gases. Adequate bottom-up estimates have been lacking, owing to the absence of reporting requirements for these gases from China and South Korea, and top-down estimates have been hampered by poor measurement coverage owing to the technical

complexities required to measure these volatile, low abundance gases at high precision. Improvements in both bottom-up information and measurement coverage, alongside refinements in transport modelling and developments in inversion methodologies, will lead to improved optimal emissions estimates of these gases in future studies.

## Acknowledgements

Observations at GSN were supported by the Basic Science Research Program through the National Research Foundation of Korea (NRF) funded by the Ministry of Science, ICT & Future Planning (2014R1A1A3051944). The UK's Department for Business, Energy & Industrial Strategy (BEIS) funded the MHD measurements and the development of InTEM.

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

**Figures**

Figure 1: Schematic of the boundary of the domain as applied in the inversion. 11 boundary
conditions were estimated as depicted from 1 to 11 as a fraction of the prior baseline
estimated using data from the Mace Head observatory. Below 6 km the boundary was divided
8 times: NNE, ENE, ESE, SSE, SSW, WSW, WNW and NNW; between 6 to 9 km the
boundary was just divided between north and south; and air arriving from above 9 km was
considered from one 'high' boundary.

Figure 2: Time series of $CF_4$ measurements during 2013 – an example year with the most
uninterrupted time series. Prior baseline (blue) is adjusted in the inversion using the baseline
condition variables, producing a posterior baseline (red). During the summer months the
proportion of air arriving from the south significantly rises causing a large shift in the
posterior baseline relative to the prior baseline calculated from Mace Head data.

Figure 3: Time series of country emission totals 2008-2015

Figure 4: The effect of the regridding routine on posterior emission distributions for $CF_4$.
Maps A, D and E are posterior emissions maps at the initial inversion resolution, at 0
regridding steps, at 25 regridding steps and at 50 regridding steps, respectively. Maps B, D, F
show the emissions magnitude minus the uncertainty calculated for each inversion grid box at
the same regridding levels (0, 25, and 50), which demonstrates the relative certainty of the
emissions distribution obtained for South Korea. Results are from inversions with initial
uncertainty on the prior emissions field is set to 100 times emissions at each fine grid square.
Units in Gg m$^{-2}$ yr$^{-1}$.

Figure 5: Emissions maps for all years of data available for $CF_4$. Results are from inversions
with initial uncertainty on the prior emissions field is set to 100 times emissions at each fine
grid square. Units in Gg m$^{-2}$ yr$^{-1}$. See Figure S7 for corresponding maps of emissions
magnitude minus the uncertainty.

Figure 6: Emissions maps for both years of data available for $NF_3$: Maps A and C are
posterior emissions maps for years 2014 and 2015, respectively. Maps B and D show the



emissions magnitude minus the uncertainty calculated for each inversion grid box maps for years 2014 and 2015, respectively. Results are from inversions with initial uncertainty on the prior emissions field is set to 100 times emissions at each fine grid square. Units in Gg m$^{-2}$ yr$^{-1}$.

Figure 7: As for Figure 5 but for HFC-23


**Tables**

Table 1: Annual posterior emissions estimates for the five main emitting countries surrounding GSN (Gg yr$^{-1}$). These posterior emissions estimates are from the inversion that uses a prior emissions uncertainty on each fine grid cell of 100x the prior emission.






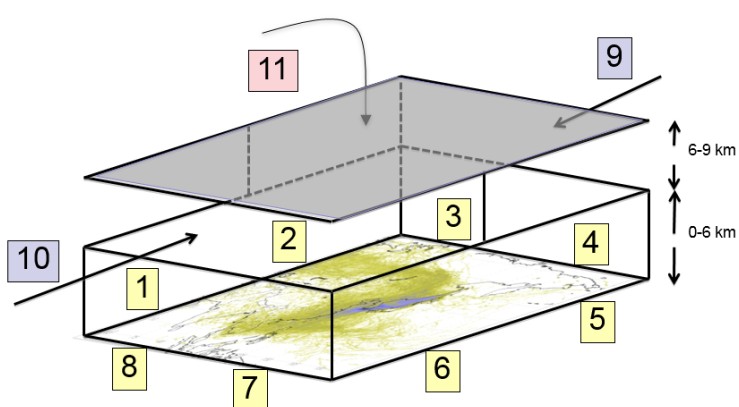

Figure 1



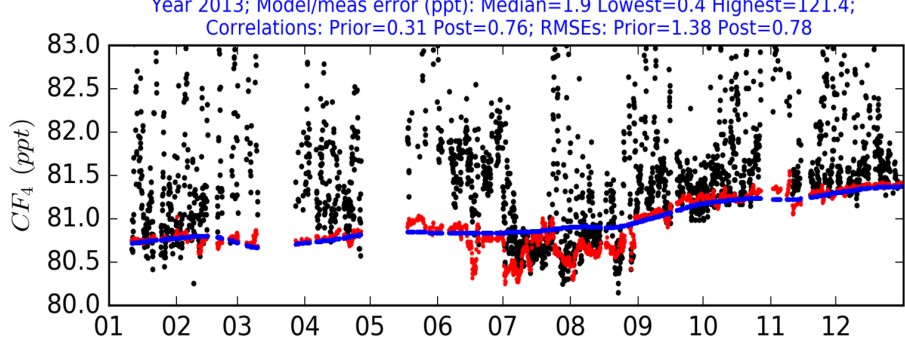

Figure 2





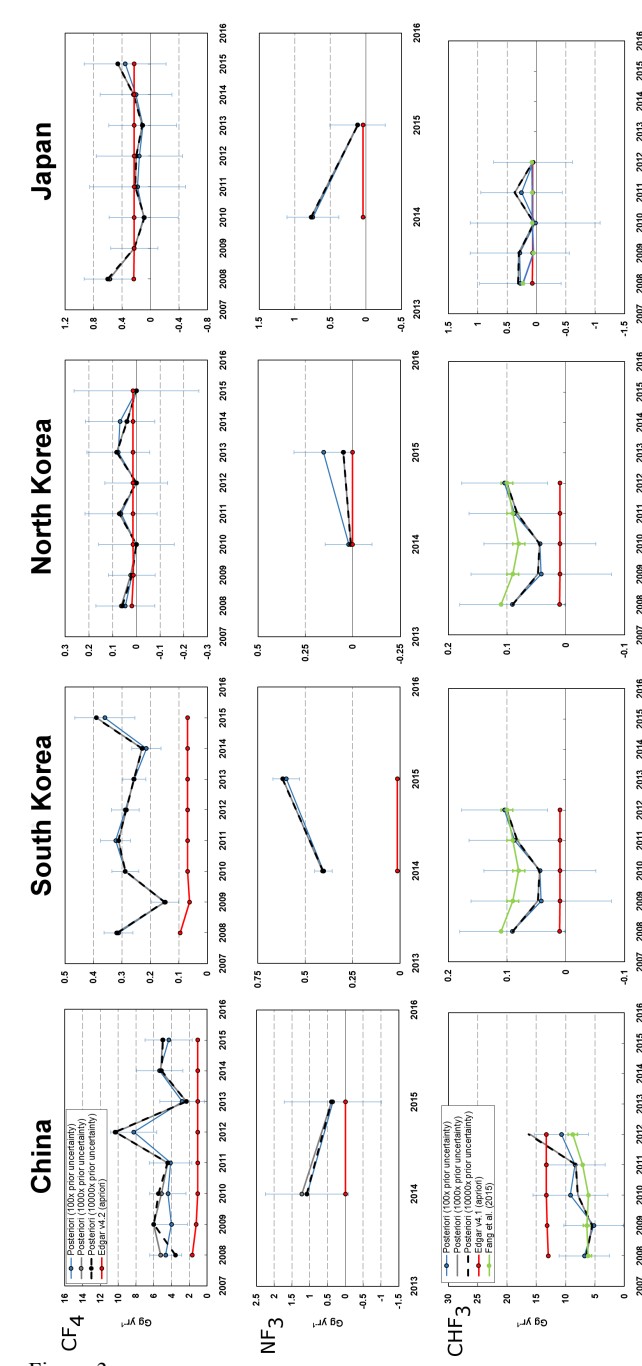

Figure 3





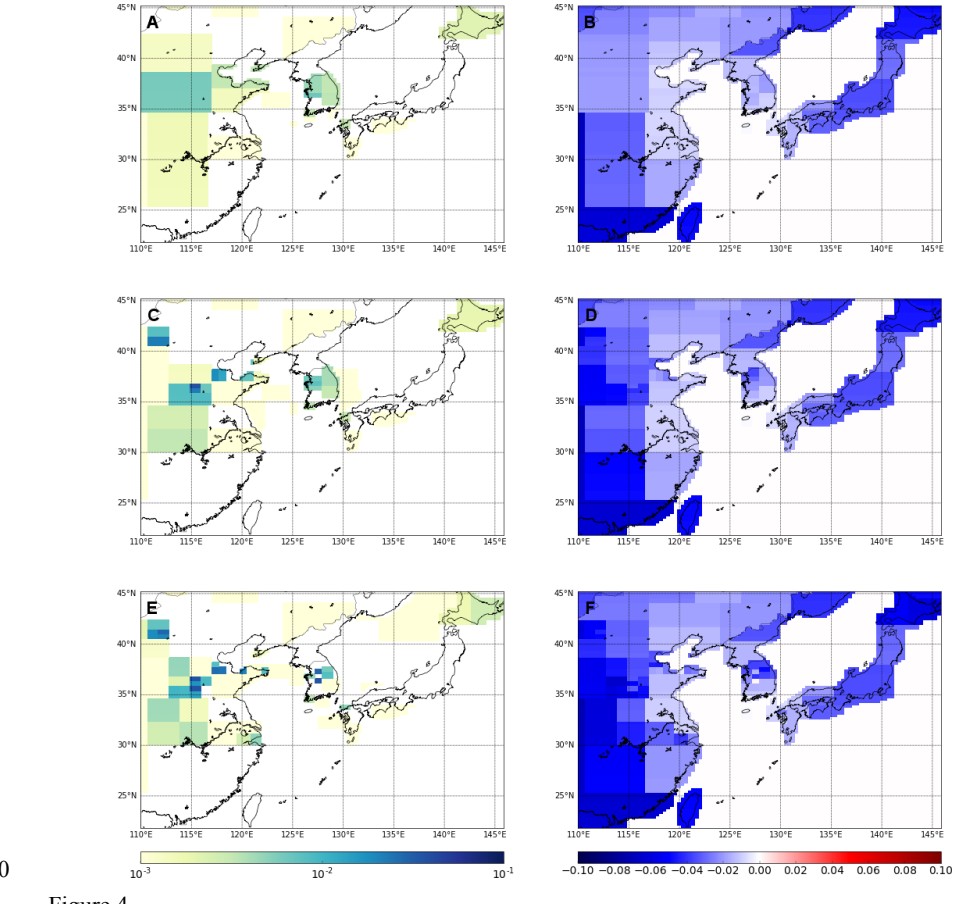


Figure 4





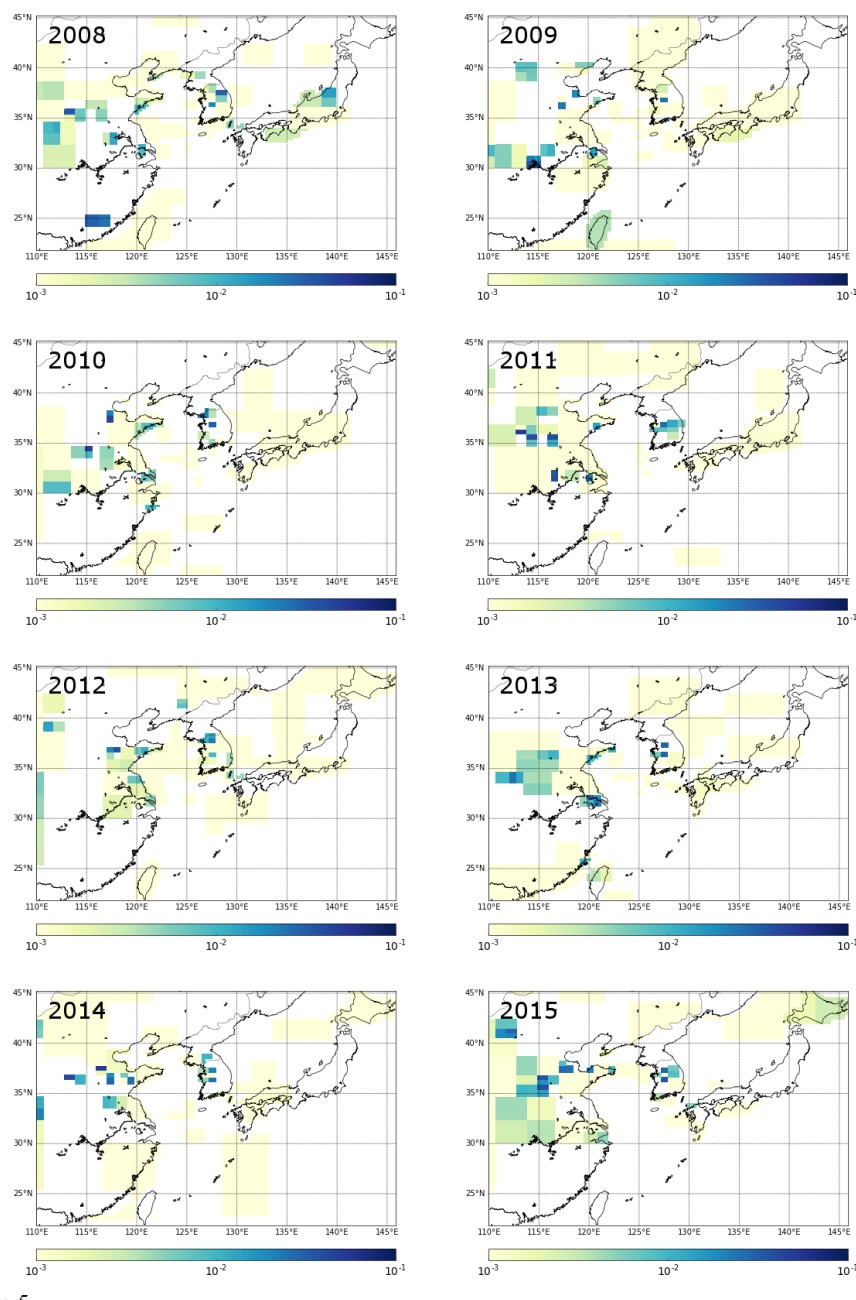

Figure 5






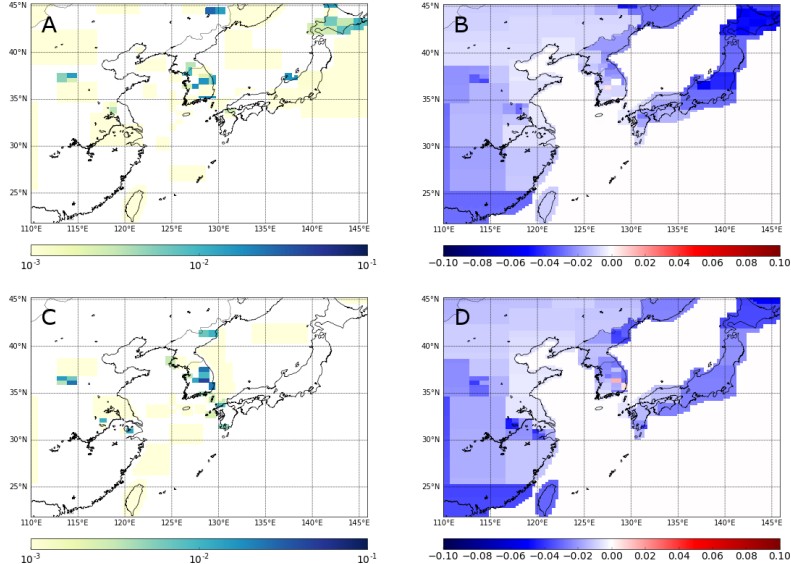

Figure: 6





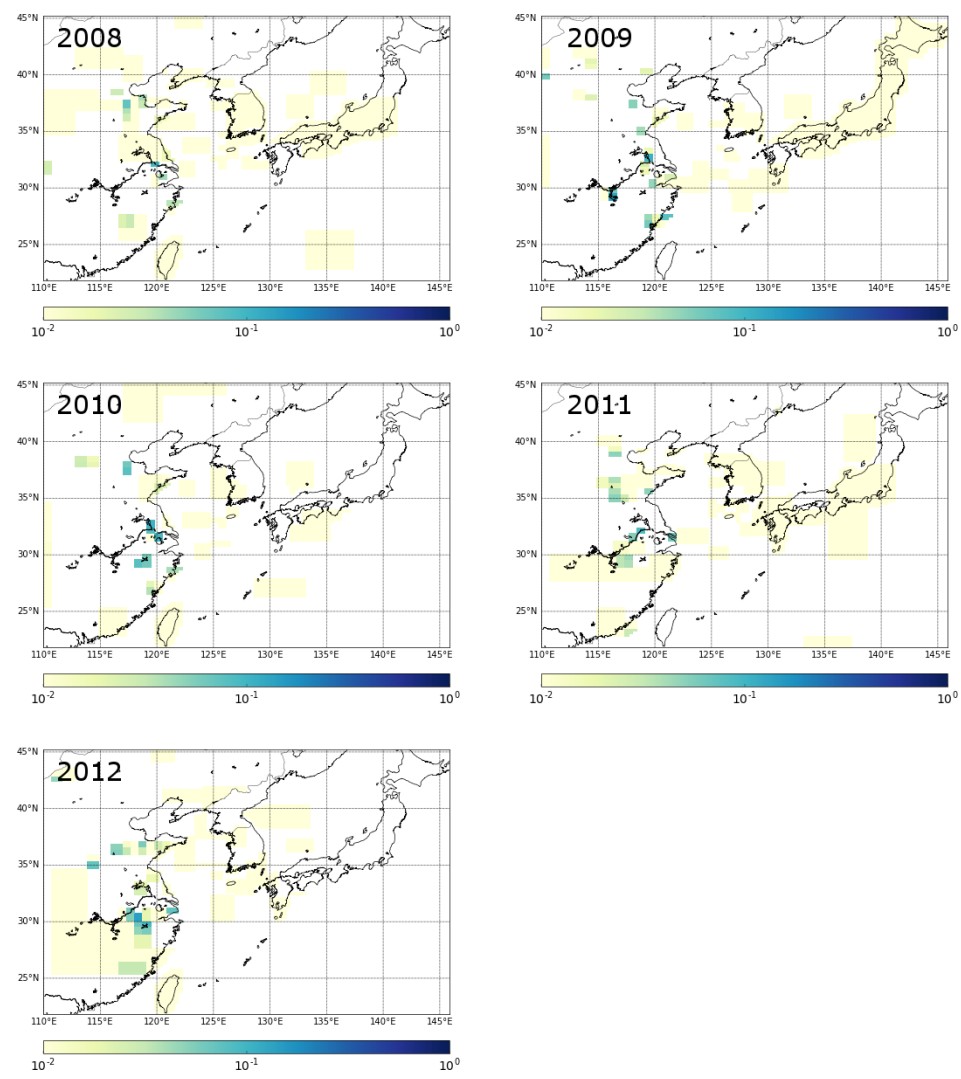


Figure 7





|      | CF4 | | | | | NF3 | | | | | HFC-23 | | | | |
|------|------|---------|---------|---------|--------|-------|---------|---------|-------|--------|--------|---------|---------|--------|--------|
|      | China | S.Korea | N.Korea | Japan | Taiwan | China | S.Korea | N.Korea | Japan | Taiwan | China | S.Korea | N.Korea | Japan | Taiwan |
| 2008 | 4.66 (1.82)# | 0.31 (0.05)# | 0.05 (0.12)# | 0.57 (0.36)# | 0.01 (0.07) | | | | | | 6.8 (4.3) | 0.09 (0.09) | 0.08 (0.28) | 0.28 (0.69) | 0.11 (0.15) |
| 2009 | 4.01 (1.80) | 0.15 (0.05) | 0.02 (0.10) | 0.23 (0.33) | 0.32 (0.17) | | | | | | 5.2 (5.1) | 0.04 (0.12) | 0.00 (0.29) | 0.29 (0.84) | 0.00 (0.48) |
| 2010 | 4.42 (2.06) | 0.29 (0.05) | 0.00 (0.16) | 0.10 (0.48) | 0.06 (0.13) | | | | | | 9.2 (6.4) | 0.04 (0.10) | 0.00 (0.39) | 0.02 (1.11) | 0.00 (0.31) |
| 2011 | 4.12 (2.37) | 0.32 (0.05) | 0.06 (0.15) | 0.18 (0.67) | 0.00 (0.26) | | | | | | 8.4 (5.1) | 0.09 (0.08) | 0.00 (0.27) | 0.26 (0.69) | 0.00 (0.41) |
| 2012 | 8.25 (2.59) | 0.29 (0.05) | 0.00 (0.13) | 0.16 (0.60) | 0.04 (0.40) | | | | | | 10.7 (4.6) | 0.10 (0.07) | 0.00 (0.23) | 0.06 (0.67) | 0.24 (0.46) |
| 2013 | 2.82 (2.49) | 0.26 (0.04) | 0.08 (0.13) | 0.11 (0.48) | 0.09 (0.26) | | | | | | | | | | |
| 2014 | 5.35 (2.61) | 0.21 (0.05) | 0.07 (0.15) | 0.21 (0.50) | 0.00 (0.30) | 1.08 (1.17) | 0.40 (0.05) | 0.02 (0.12) | 0.75 (0.36) | 0.03 (0.09) | | | | | |
| 2015 | 4.33 (2.65) | 0.36 (0.11) | 0.00 (0.26) | 0.36 (0.57) | 0.00 (0.44) | 0.36 (1.36) | 0.60 (0.07) | 0.15 (0.16) | 0.11 (0.39) | 0.00 (0.27) | | | | | |

# Kim et al. (2010) estimated CF$_4$ emissions from China in the range 1.7-3.1 Gg yr$^{-1}$ and Li et al. (2011) 1.4-2.9 Gg yr$^{-1}$. For South and North Korea (combined) Li et al. (2011) estimated emissions of CF$_4$ at 0.19-0.26 Gg yr$^{-1}$ and from Japan at 0.2-0.3 Gg yr$^{-1}$.

Table