# Peer review of "Inverse modelling of CF4 and NF3 emissions in East Asia"

_Atmospheric Chemistry and Physics, 2017_

## Referee Comment (RC1) · A. Stohl (Referee) · 24 Feb 2018

This is a paper describing an inverse modelling study for the greenhouse gases CF4 and NF3, which are particularly worrisome because of their long lifetimes. These are rarely measured gases with few existing (and quite unreliable) regional emission estimates. Therefore, this study adds important new information on emissions in East Asia. The methods are largely solid and I recommend publication. I have, however, a number of comments that I would like the authors to consider before final publication.

Major

I am somewhat concerned by the large interannual variability of national emissions obtained by the inversion. For example, in Table 1, CF4 emissions in China in 2012

are 8.25 Gg/year, but in 2013, they are only 2.82 Gg/year. Is a 65% reduction from one year to the other realistic? This is true also for other species (e.g., NF3 changes from 1.08 Gg/year to 0.36 Gg/year from 2014 to 2015) and partly also for other countries. For the first example, the change is also outside the combined uncertainty range. I think this needs at least some discussion. How do the authors interpret this? As an inversion artifact? Real changes?

Abstract, line 25: The sentence "Owing to the poor availability of good prior information for this study our results are strongly constrained by the atmospheric measurements." is a wrong statement. The constraint offered by the measurements does not become any better with weak prior information. Of course, relatively speaking, the measurements get more weight in the inversion, but that doesn't mean that the constraint is strong. It also does not mean that uncertainties are lower than with better prior information. In fact, your constraints are very weak for most regions (as you yourself repeatedly point out), which is a consequence of using only data from one station.

The fact that data from only one station were used is problematic in itself. I am aware that there are not many stations measuring CF4 and NF3 but was there nothing at all available (e.g., Japanese data)? In my experience, inversions using data from only one site are not very "stable" and the large interannual variability in country-total emissions obtained seems to confirm this.

Line 171+ Lines 183-184: air concentration (dosage), units: The units used for the NAME backward runs don't make sense to me. How can a mass be emitted in backward mode, and how can concentrations be obtained as output? The output should be a sensitivity of receptor concentrations (or, alternatively, mixing ratios) to emission fluxes (either per grid cell, or by square metre, or by volume, possibly by time). But not concentrations.

Equation 3 and text describing it: Why do you include f_topography and f_inlet, if they are anyway always 1? For the sake of both brevity and clarity, this should be removed.

It seems to be something in development that is not yet used. Further, it's a good idea to try quantifying observation/model uncertainty, as this is certainly not a constant (as often assumed in other studies). However, the model uncertainty is not only determined by the boundary layer height at the receptor. Equally important is the boundary layer height in the source regions, and of course there are many other factors determining model uncertainty. Can you discuss/justify/explain your choice of uncertainty scaling, without forgetting to mention that there are many other factors influencing uncertainty?

Paragraph starting at line 318 (and paragraph before): I think it would be good if you could put the scaling factors (posteriors) which you obtain for the MH baseline into a table, or perhaps better even, add the numbers to Figure 1? I assume these are constant values? Or are they allowed to vary with time?

Figure 4: What exactly does the right column show? If I understand right it is the emission flux minus its uncertainty (but I might be wrong, this needs better explanation). But does this quantity make sense? It's the lower estimate of the flux, and all values seem to be negative? Here, I would normally expect something like a map of the uncertainty reduction (e.g., in %) due to the inversion (perhaps does not make so much sense in your case, as prior uncertainties are kind of arbitrary), or a map of the uncertainty itself.

Minor

Lines 44+443: NF3, which contains no carbon, does not contribute to carbon budgets. This needs rephrasing, it's only CO2 equivalents, which you mean.

Line 46: Isn't it misleading to compare such long-lived species to CO2 using GWP100? The lifetime impact of CF4 is orders of magnitude higher than GWP100 suggests.

Line 86: production areas: maybe better production sites?

Lines 200-201: I don't understand what you mean with "the sensitivity of changes in boundary conditions to measured mixing ratios".

Line 204-205: You need to say which numerical method you have used for cost function minimization. "Non-negative least squares fit" is not concrete enough.

Line 271: What is meant with "If the model boundary layer height transitions across the sample inlet height"? In particular, what is meant with "transitions"? Is it above, or below, or equal, or what? Is there some time development involved, as the word suggests?

Line 276: Equation is not numbered.

Lines 332-334: The sentence "Further, although the very large area for our model domain may not be necessary for this study, the model will not need to be run again should a larger area of analysis be required." does not add any information to this paper. I would suggest removing it.

Line 364-365, "For estimates of emissions from other countries our posterior estimates are not constrained sufficiently to make a meaningful comparison with Fang et al. (2015)." OK, but yet you show the comparison in Figure 3. This is a bit contradictory.

Paragraph starting at line 487: Reference to Figure 7 is missing.

Line 697: "Maps A, D and E"... D should be C, I think.

Typos, etc.

Line 114: able TO measure

Line 364: Stohl et al. (2015) should be Stohl et al. (2010).

---

## Referee Comment (RC2) · Anonymous Referee #2 · 11 Mar 2018

The authors discuss measurements of CF4 and NF3, along with HFC-23, in South-East Asia and derive country wide emissions of these compounds. This is very relevant work considering the large global warming potential of these compounds and increasing use. The paper is well written, scientifically sound and has a clear structure. The technical descriptions are not easy to understand and maybe could be improved by including a figure showing the region of influence of emissions on the measurement site. The figures could also be improved by adding labels and headers so they are more easily understood without reading the caption. I think the paper is suitable for ACP. Below are a number of smaller comments that could improve the paper.

L18: I don't understand the 'Well mixed' before 'abundances'. What do you want to convey? L29: Remove digits here: 4.3 +/- 2.7 Gg L24-26: Why do you mention the

poor prior in the abstract? Also with a good prior the derived results would hopefully depend on the measurements and not on the prior information. L30: Add a digit to 0.6 to be in agreement with the 0.07. L60: What is the reason the GWP is estimated higher? Because of lifetime? Please add this here. L108: Add here that developing countries are not required to report. L117: Add where Jeju Island is located, other ways the rest of the sentence cannot be placed. I also suggest to add the marker at the GSN station in Figure 4, 5, and 6 and refer here to this figures for reference. L185ff: It would be very useful if a figure could be included showing matrix D. This would make it clear the regions the inversion is most sensitive for. L319: I assume MH should be NH. L494: Is 'remove' the correct word here? The influence of the prior is reduced and replaced by the observational data. L697: Typo: Maps A, C and E, not A, D, and E. L740, Figure 4: The readability would be greatly improved if labels/headers are added, instead of explaining all in the caption. Figure 6: Same comments as for Figure 4.

---

## Author Response (AR1)

Author response to both reviews follows. A copy of the reviewer comment is given (with bullet point '-') followed by a response.

Response to reviewer 1

- This is a paper describing an inverse modelling study for the greenhouse gases $CF_4$ and NF3, which are particularly worrisome because of their long lifetimes. These are rarely measured gases with few existing (and quite unreliable) regional emission estimates. Therefore, this study adds important new information on emissions in East Asia. The methods are largely solid and I recommend publication. I have, however, a number of comments that I would like the authors to consider before final publication.

We thank Reviewer 1 (A Stohl) for the constructive feedback.

- Major
- I am somewhat concerned by the large interannual variability of national emissions obtained by the inversion. For example, in Table 1, CF4 emissions in China in 2012 are 8.25 Gg/year, but in 2013, they are only 2.82 Gg/year. Is a 65% reduction from one year to the other realistic? This is true also for other species (e.g., NF3 changes from 1.08 Gg/year to 0.36 Gg/year from 2014 to 2015) and partly also for other countries. For the first example, the change is also outside the combined uncertainty range. I think this needs at least some discussion. How do the authors interpret this? As an inversion artifact? Real changes?

The reviewer is correct to highlight our lack of discussion on the interannual variations observed in Figure 3. Our revised manuscript will include a more thorough discussion of interannual changes, also discussed in the following points.
Our East Asia regional emission estimates can be compared to global totals from global atmospheric monitoring sites. Rigby et al.[1] estimated global $CF_4$ emissions of 10.4±0.6 Gg/year in 2008 with a steady but small increase to 11.1 ± 0.4 Gg/year in 2013 (with the exception of a dip in 2009 to 9.3±0.5 Gg/year). We highlight that our Chinese emission estimates remain within a narrow range for 5 of the 8 years studied at between 4.0 and 4.7 Gg/year (with typical uncertainties <2.7 Gg/year), and for 7 of the 8 years studied between 2.82 and 5.35 Gg/year. However, the estimate for 2012 appears to be anomalous at 8.25±2.59 Gg/year. In relation to the global top-down estimates from 2008 to 2012, our Chinese estimates represent between 37 to 45 % of global emissions between 2008 and 2011 with a jump to 74 % in 2012. Thus, this significant increase in 2012 is not reconcilable with atmospheric measurements on the global scale and is very likely a spurious result of the inversion.

The most probable explanation for such a result is the incorrect assignment of emissions on the inversion grid. Incorrect assignment of emissions can occur between countries, particularly where air parcels frequently pass over more than one country, therefore reducing the ability of the inversion to confidently place emissions. However, there is not an obvious drop in emissions for another country in 2012 that would offset the large increase in the Chinese emissions estimate. Within a country, incorrect assignment of emissions from an area closer to the receptor to an area further from the receptor will increase the calculated total emissions owing to increased dilution in going from a near to a far source. Our inversion is susceptible to this effect as we only have one site for assimilation of measurements; two measurement sites, spaced apart and straddling the area of interest, would provide significantly more information to constrain the spatial emissions distribution.

We argue that other large changes in our emissions estimates could be real. For example, Japan's National Inventory Report for $NF_3$ show a reduction in emissions of 73% between 2013 and 2015, which is within the uncertainty of the relative rate of decrease we observe.

- Abstract, line 25: The sentence "Owing to the poor availability of good prior information for this study our results are strongly constrained by the atmospheric measurements" is a wrong statement. The constraint offered by the measurements does not become any better with weak prior information. Of course, relatively speaking, the measurements get more weight in the inversion, but that doesn't mean that the constraint is strong. It also does not mean that uncertainties are lower than with better prior information. In fact, your constraints are very weak for most regions (as you yourself repeatedly point out), which is a consequence of using only data from one station. The fact that data from only one station were used is problematic in itself. I am aware that there are not many stations measuring CF4 and NF3 but was there nothing at all available (e.g., Japanese data)? In my experience, inversions using data from only one site are not very "stable" and the large interannual variability in country-total emissions obtained seems to confirm this.

We acknowledge the reviewer's concerns here in our choice of wording. Instead of 'strongly constrained' we will say 'Owing to the poor availability of good prior information for this study our emissions estimates are largely influenced by the atmospheric measurements'. Unfortunately, $NF_3$ is currently not measured at any other site in East Asia, and $CF_4$ has very sparse datasets with poor availability of information regarding data quality.

- Line 171+ Lines 183-184: air concentration (dosage), units: The units used for the NAME backward runs don't make sense to me. How can a mass be emitted in backward mode, and how can concentrations be obtained as output? The output should be a sensitivity of receptor concentrations (or, alternatively, mixing ratios) to emission fluxes (either per grid cell, or by square metre, or by volume, possibly by time). But not concentrations.

Inert particles are released backwards which are acted on by the transport model, however, NAME associates a mass to these trajectories. Hence, NAME output is provided as the time integrated surface concentration ($g \ s \ m^{-3}$) in each grid cell – the surface influence resulting from a conceptual release at a rate of 1 g $s^{-1}$ from the site. 'Offline' this surface influence is divided by the total mass emitted during the 1-hour release time and multiplied by the geographical area of each grid box to form a new array with each component representative of how 1 g $m^{-2}$ $s^{-1}$ of continuous emissions from a grid square would result in a measured concentration at the model's release point. Multiplication of each grid component by an emission rate would then result in a contribution to the concentration.
It's convenient to design NAME in this way as source and output routines can be identical for forwards and backwards calculations, and although the hypothetical concentration values are not what is wanted, it is straight forward to convert these to sensitivities as explained above.

We will include this description and explanation in section 2.2 in the revision.

- Equation 3 and text describing it: Why do you include f_topography and f_inlet, if they are anyway always 1? For the sake of both brevity and clarity, this should be removed. It seems to be something in development that is not yet used. Further, it's a good idea to try quantifying observation/model uncertainty, as this is certainly not a constant (as often assumed in other studies). However, the model uncertainty is not only determined by the boundary layer height at the receptor. Equally important is the boundary layer height in the source regions, and of course there are many other factors determining model uncertainty. Can you discuss/justify/explain your choice of uncertainty scaling, without forgetting to mention that there are many other factors influencing uncertainty?

We will remove f_topography and f_inlet from the equation as they do not affect our uncertainty calculation for the Gosan site.  We will include a discussion on why we use model boundary layer height at the receptor as a way to capture model uncertainty.
A low boundary layer (causing a larger model uncertainty) has two implications for measurements at the Gosan site:

1) A greater possibility of air from above the boundary layer being sampled in reality but not in the model. Subtle changes in the boundary layer height at the exact measurement location are not well modelled and the difference between sampling above or below the boundary layer can have a significant influence on the amount of pollutant assigned to a back trajectory.

2) Greater influence of emissions from sources very near Gosan. A lower boundary layer means that a lower rate of dilution of local emissions will occur, in turn increasing the signal of the local pollutant above the baseline. A relatively small change in a low boundary layer will have a significant influence on this dilution compared to the same change on a high boundary layer. Thus, any error in the boundary layer height at low levels can significantly amplify the uncertainty in the pollutant dilution. This is coupled with the fact that the modelled boundary layer has significant uncertainty especially when low.

We agree that all elements of the modelled meteorology are important in understanding the dilution and uncertainty in modelling from source to receptor. Ideally you would need to understand the uncertainty and impact of each element of meteorology (wind speed and direction, boundary layer height, temperature, pressure, etc) that a model particle experiences in order to fully quantify the model uncertainty at each measurement time. At this current time this is significantly beyond what is available from numerical weather prediction models. So in order to attempt to quantify a model/observation uncertainty we have had to take the pragmatic view and use modelled boundary layer height at the receptor as a proxy, but we agree that that it is very simplistic. Moving forward, concentration and meteorology measurements at multiple heights at the receptor would be a significant improvement.  Undertaking such a boundary layer analysis at every possible emission location would be very difficult, especially given that the time that each particle has reached each part of the ground is not recorded in our NAME runs.
We will include a more thorough discussion and explanation of our approach in a revision.

- Paragraph starting at line 318 (and paragraph before): I think it would be good if you could put the scaling factors (posteriors) which you obtain for the MH baseline into a table, or perhaps better even, add the numbers to Figure 1? I assume these are constant values? Or are they allowed to vary with time?

We will add these values to Figure 1 for $CF_4$ for 2013, which matches the example time series output of Figure 2. We will also make clear in the text that these scaling factors are allowed to vary annually (alongside the emissions).

- Figure 4: What exactly does the right column show? If I understand right it is the emission flux minus its uncertainty (but I might be wrong, this needs better explanation). But does this quantity make sense? It's the lower estimate of the flux, and all values seem to be negative? Here, I would normally expect something like a map of the uncertainty reduction (e.g., in %) due to the inversion (perhaps does not make so much sense in your case, as prior uncertainties are kind of arbitrary), or a map of the uncertainty itself.

As the reviewer highlights, the common way to illustrate grid-level uncertainty is in an 'uncertainty reduction' map. This works well when starting from a relatively well-constrained, spatially resolved prior to illustrate the what additional constraint the atmospheric observations bring; however, in this study we are starting from very poor prior information and we generate a posterior emission map that is very distinct from the prior, informed largely by the measurements. Thus, an uncertainty reduction map provides little useful information.

Our aim with the emissions-minus uncertainty maps is to provide information on where we are most certain of large emissions i.e. where emission hotpots are located and if they are significant: Less negative values indicate more certainty, with positive values indicating that the uncertainty is less than the best estimate and negative values indicating that the uncertainty is bigger than the estimate. Positive values can be observed e.g. Figure 6D. A map of the uncertainties alone would not provide any information relating to emissions significance, and percentage uncertainty does not give information distinguishing emission magnitudes.

As this is a different way of illustrating uncertainties on a grid, we will ensure to justify and explain our approach in a revised text.

- Minor
- Lines 44+443: NF3, which contains no carbon, does not contribute to carbon budgets. This needs rephrasing, it's only CO2 equivalents, which you mean.

We will rephrase this to remove any confusion

- Line 46: Isn't it misleading to compare such long-lived species to CO2 using GWP100? The lifetime impact of CF4 is orders of magnitude higher than GWP100 suggests.

We are trying to relate the calculated emission using a policy metric, and not addressing the climate impact of these gases over different time horizons. The use of GWP-100 to compare gases is an imperfect but widely used method. We will address the limitations of the GWP metric in our revised manuscript.

- Line 86: production areas: maybe better production sites?

Agree – this will be changed

- Lines 200-201: I don't understand what you mean with "the sensitivity of changes in boundary conditions to measured mixing ratios".

'Boundary conditions' is a confusing term given the similarity with the 'boundary layer'. We will replace this terminology with 'domain edge'. Included in $H$ is the sensitivity of mixing ratios to changes in the boundary edge multiplying factors.

- Line 204-205: You need to say which numerical method you have used for cost function minimization. "Non-negative least squares fit" is not concrete enough.

We will write in the paper: We use the "NNLS" (non-negative least squares) algorithm of Lawson and Hanson (1974) for finding the least squares fit under the constraint that the emissions are non-negative. This is an "active set" method which efficiently iterates over choices for the set of emissions for which the non-negative constraint is active, i.e. the set of emissions which are set to zero.

- Line 271: What is meant with "If the model boundary layer height transitions across the sample inlet height"? In particular, what is meant with "transitions"? Is it above, or below, or equal, or what? Is there some time development involved, as the word suggests.

We will correct the phrasing to: "If the model boundary layer height is close to the sample inlet height (i.e. less certainty on whether sampling is above or below the boundary layer) then…."

-   Line 276: Equation is not numbered.

As we do not refer to this equation in the text we don't believe this needs to be numbered.

    -   Lines 332-334: The sentence "Further, although the very large area for our model domain may not be necessary for this study, the model will not need to be run again should a larger area of analysis be required." does not add any information to this paper. I would suggest removing it.

Agreed, we will remove this.

    -   Line 364-365, "For estimates of emissions from other countries our posterior estimates are not constrained sufficiently to make a meaningful comparison with Fang et al. (2015)." OK, but yet you show the comparison in Figure 3. This is a bit contradictory.

We'll remove this sentence to avoid any contradiction.

    -   Paragraph starting at line 487: Reference to Figure 7 is missing.

We will add this

-   Line 697: "Maps A, D and E": : : D should be C, I think.

Yes, corrected

    -   Typos, etc.
-   Line 114: able TO measure
    -   Line 364: Stohl et al. (2015) should be Stohl et al. (2010).

Corrected

Response to reviewer 2

- The authors discuss measurements of CF4 and NF3, along with HFC-23, in South-
East Asia and derive country wide emissions of these compounds. This is very
relevant work considering the large global warming potential of these compounds
and increasing use. The paper is well written, scientifically sound and has a clear
structure. The technical descriptions are not easy to understand and maybe could be
improved by including a figure showing the region of influence of emissions on the
measurement site. The figures could also be improved by adding labels and headers
so they are more easily understood without reading the caption.

We will improve the annotation of the figures to help improve the ease with which they can
be understood.

- I think the paper is suitable for ACP. Below are a number of smaller comments that
could improve the paper. L18: I don't understand the 'Well mixed' before
'abundances'. What do you want to convey?

We will replace the first sentence with: "Decadal trends in the atmospheric abundances of…"

- L29: Remove digits here: 4.3 +/- 2.7 Gg L24-26:

We will alter this.

- Why do you mention the poor prior in the abstract? Also with a good prior the derived
results would hopefully depend on the measurements and not on the prior
information.

Resolved under previous review

- L30: Add a digit to 0.6 to be in agreement with the 0.07.
We will alter this.

- L60: What is the reason the GWP is estimated higher? Because of lifetime? Please
add this here.
This is due to a change in estimate of the radiative efficiency of NF3

- L108: Add here that developing countries are not required to report.
We would prefer to leave this as it is rather than using developing and developed wording for
the countries listed under Annex 1, Annex 2

- L117: Add where Jeju Island is located, other ways the rest of the sentence cannot
be placed. I also suggest to add the marker at the GSN station in Figure 4, 5, and 6
and refer here to this figures for reference.
We will add Jeju Island, Republic of Korea, and add a marker for the station in the relevant
figures.

- L185ff: It would be very useful if a figure could be included showing matrix D. This
would make it clear the regions the inversion is most sensitive for.
We will add a NAME aggregated sensitivity map to illustrate the spatial sensitivity of the
inversion.

- L319: I assume MH should be NH.
This should be 'MHD' for Mace Head – this will be corrected.

-  L494: Is 'remove' the correct word here? The influence of the prior is reduced and replaced by the observational data.

We revise our wording of this topic throughout the manuscript based on the comments of the previous reviewer. Here we will replace "We largely remove.." with "We reduce.."

-  L697: Typo: Maps A, C and E, not A, D, and E.

Corrected

       -  L740, Figure 4: The readability would be greatly improved if labels/headers are added, instead of explaining all in the caption. Figure 6: Same comments as for
 Figure 4.

As explained above we will improve the annotation of our emissions maps

Additional references

[1] Rigby, M., et al., Recent and future trends in synthetic greenhouse gas radiative
forcing. *Geophys. Res. Lett.* **2014,** *41* (7), 2623-2630.

**Inverse modelling of $CF_4$ and $NF_3$ emissions in East Asia**

Tim Arnold[1,2,3*], Alistair J. Manning[3], Jooil Kim[4], Shanlan Li[5], Helen Webster[3], David Thomson[3], Jens Mühle[4], Ray F. Weiss[4], Sunyoung Park[5,6], and Simon O'Doherty[7]

[1]National Physical Laboratory, Teddington, Middlesex, UK

[2]School of GeoSciences, University of Edinburgh, Edinburgh, UK

[3]Met Office, Exeter, UK

[4]Scripps Institution of Oceanography, University of California, San Diego, La Jolla, California 92037, USA

[5]Kyungpook Institute of Oceanography, Kyungpook National University, Daegu 41566, Republic of Korea

[6]Department of Oceanography, Kyungpook National University, Daegu 41566, Republic of Korea

[7]School of Chemistry, University of Bristol, Bristol, UK.

*Corresponding author tim.arnold@ed.ac.uk

Decadal trends in the atmospheric abundances of carbon tetrafluoride ($CF_4$) and nitrogen trifluoride ($NF_3$) have been well characterised and have provided a time series of global total emissions. Information on locations of emissions contributing to the global total, however, is currently poor. We use a unique set of measurements between 2008 and 2015 from the Gosan station, Jeju Island, South Korea (part of the Advanced Global Atmospheric Gases Experiment network), together with an atmospheric transport model to make spatially disaggregated emission estimates of these gases in East Asia. Owing to the poor availability of good prior information for this study our emissions estimates are largely influenced by the atmospheric measurements. Notably, we are able to highlight emissions hotspots of $NF_3$ and $CF_4$ in South Korea, owing to the measurement location. We calculate emissions of $CF_4$ to be quite constant between years 2008 and 2015 for both China and South Korea with 2015 emissions calculated at $4.3 \pm 2.7$ Gg yr$^{-1}$ and $0.36 \pm 0.11$ Gg yr$^{-1}$, respectively. Emission estimates of $NF_3$ from South Korea could be made with relatively small uncertainty at $0.6 \pm$

Commented [TA1]: We have changed our wording to reflect comments by reviewer 1 (and reviewer 2). This change is reflected in the rest of the text.

[revised manuscript text omitted]

Commented [TA2]: We have improved our explanation of how NAME is set up, and how the model output is worked up into a dilution matrix, based on reviewer 1's comments.

[revised manuscript text omitted]

**Commented [TA3]:** Following Reviewer 1's comments we now explain and justify our approach to handling model uncertainty.

lower rate of dilution of local emissions will occur, in turn increasing the signal of the local pollutant above the baseline. A relatively small change in a low boundary layer will have a significant influence on this dilution compared to the same change on a high boundary layer. Thus, any error in the boundary layer height at low levels can significantly amplify the uncertainty in the pollutant dilution. This is coupled with the fact that the modelled boundary layer has significant uncertainty especially when low.

To assign a model uncertainty to each hourly window of measurements we use model information of BLH:

$$\sigma_{model} = \sigma_{baseline} \times f_{BLH} \quad (3)$$

where, $\sigma_{baseline}$ is the variability associated with the baseline calculation (see Section 2.6), and $f_{BLH}$ is a multiplying factor (greater than or less than unity) that increases or decreases the relative uncertainty assigned to each model time period. $f_{BLH}$ is based on modelled boundary layer height magnitude and variability over a three-hour period and is calculated with the following:

$$f_{BLH} = \frac{Max_{BLH-inlet}}{Min_{BLH-inlet}} \times \frac{Threshold}{Min_{BLH}}$$

where, $Max_{BLH-inlet}$ is the largest of either 100 m or the maximum distance, calculated hourly, between the inlet and the modelled BLH within a period of three hours around the measurement time; $Min_{BLH-inlet}$ is the smallest of the distances calculated between the inlet and the BLH over the same three-hour period; $Threshold$ is an arbitrary value set at 500 m; and $Min_{BLH}$ is the lowest boundary layer height recorded over the three-hour period. Thus, the relative assigned uncertainty takes into account the proximity of the varying boundary layer to the inlet height and a recognition that observations taken when the boundary layer is varying at higher altitudes (>500 m a.g.l.) is likely to have less impact and therefore have lower uncertainty compared to those taken when the BLH is varying at lower altitudes (< 500 magl).

Supporting Figures S2-S6 show annual time series of observations and the corresponding measurement-model uncertainties, as well as statistics for the mismatch between observations and modelled time series.

**2.6 Baseline calculation and domain border conditions**

Commented [TA4]: As suggested by reviewer 1 we have removed the other multiplying factors that are not of relevance to our study

[revised manuscript text omitted]

Commented [TA5]: We have added a discussion on the annual variability in emissions estimates following comment by reviewer 1.

[revised manuscript text omitted]

Maximum value = 16.067 s/m

[Figure]

0.003  0.010  0.030  0.100  0.300  1.000  3.000  10.000  30.000

Figure 1

Commented [TA9]: As advised to include by reviewer 2

[Figure]

Figure 2

[Figure]

Year 2013; Model/meas error (ppt): Median=1.9 Lowest=0.4 Highest=121.4;
Correlations: Prior=0.31 Post=0.76; RMSEs: Prior=1.38 Post=0.78

Figure 3

[Figure]

Figure 4

[Figure]

Figure 5

Commented [TA10]: Figure has been annotated to improve readability as advised by reviewer 2

[Figure]

Figure 6

[Figure]

Figure: 7

[Figure]

Figure 8

|  | CF$_4$ | | | | | NF$_3$ | | | | | HFC-23 | | | | |
|---|---|---|---|---|---|---|---|---|---|---|---|---|---|---|---|
|  | China | S.Korea | N.Korea | Japan | Taiwan | China | S.Korea | N.Korea | Japan | Taiwan | China | S.Korea | N.Korea | Japan | Taiwan |
| 2008 | 4.66 | 0.31 | 0.05 | 0.57 | 0.01 | | | | | | 6.8 | 0.09 | 0.08 | 0.28 | 0.11 |
| | (1.82)[#] | (0.05)[#] | (0.12)[#] | (0.36)[#] | (0.07) | | | | | | (4.3) | (0.09) | (0.28) | (0.69) | (0.15) |
| 2009 | 4.01 | 0.15 | 0.02 | 0.23 | 0.32 | | | | | | 5.2 | 0.04 | 0.00 | 0.29 | 0.00 |
| | (1.80) | (0.05) | (0.10) | (0.33) | (0.17) | | | | | | (5.1) | (0.12) | (0.29) | (0.84) | (0.48) |
| 2010 | 4.42 | 0.29 | 0.00 | 0.10 | 0.06 | | | | | | 9.2 | 0.04 | 0.00 | 0.02 | 0.00 |
| | (2.06) | (0.05) | (0.16) | (0.48) | (0.13) | | | | | | (6.4) | (0.10) | (0.39) | (1.11) | (0.31) |
| 2011 | 4.12 | 0.32 | 0.06 | 0.18 | 0.00 | | | | | | 8.4 | 0.09 | 0.00 | 0.26 | 0.00 |
| | (2.37) | (0.05) | (0.15) | (0.67) | (0.26) | | | | | | (5.1) | (0.08) | (0.27) | (0.69) | (0.41) |
| 2012 | 8.25 | 0.29 | 0.00 | 0.16 | 0.04 | | | | | | 10.7 | 0.10 | 0.00 | 0.06 | 0.24 |
| | (2.59) | (0.05) | (0.13) | (0.60) | (0.40) | | | | | | (4.6) | (0.07) | (0.23) | (0.67) | (0.46) |
| 2013 | 2.82 | 0.26 | 0.08 | 0.11 | 0.09 | | | | | | | | | | |
| | (2.49) | (0.04) | (0.13) | (0.48) | (0.26) | | | | | | | | | | |
| 2014 | 5.35 | 0.21 | 0.07 | 0.21 | 0.00 | 1.08 | 0.40 | 0.02 | 0.75 | 0.03 | | | | | |
| | (2.61) | (0.05) | (0.15) | (0.50) | (0.30) | (1.17) | (0.05) | (0.12) | (0.36) | (0.09) | | | | | |
| 2015 | 4.33 | 0.36 | 0.00 | 0.36 | 0.00 | 0.36 | 0.60 | 0.15 | 0.11 | 0.00 | | | | | |
| | (2.65) | (0.11) | (0.26) | (0.57) | (0.44) | (1.36) | (0.07) | (0.16) | (0.39) | (0.27) | | | | | |

**Kim et al. (2010) estimated CF$_4$ emissions from China in the range 1.7-3.1 Gg yr$^{-1}$ and Li et al. (2011) 1.4-2.9 Gg yr$^{-1}$. For South and North Korea (combined) Li et al. (2011) estimated emissions of CF$_4$ at 0.19-0.26 Gg yr$^{-1}$ and from Japan at 0.2-0.3 Gg yr$^{-1}$.**

Table